# Asynchronous Federated Learning System Based on Permissioned Blockchains

**DOI:** 10.3390/s22041672

**Published:** 2022-02-21

**Authors:** Rong Wang, Wei-Tek Tsai

**Affiliations:** Digital Society & Blockchain Laboratory, Beihang University, Beijing 100191, China; tsai7@yahoo.com

**Keywords:** asynchronous federated learning, permissioned blockchains, privacy protection, IoT, multi-blockchains architecture

## Abstract

The existing federated learning framework is based on the centralized model coordinator, which still faces serious security challenges such as device differentiated computing power, single point of failure, poor privacy, and lack of Byzantine fault tolerance. In this paper, we propose an asynchronous federated learning system based on permissioned blockchains, using permissioned blockchains as the federated learning server, which is composed of a main-blockchain and multiple sub-blockchains, with each sub-blockchain responsible for partial model parameter updates and the main-blockchain responsible for global model parameter updates. Based on this architecture, a federated learning asynchronous aggregation protocol based on permissioned blockchain is proposed that can effectively alleviate the synchronous federated learning algorithm by integrating the learned model into the blockchain and performing two-order aggregation calculations. Therefore, the overhead of synchronization problems and the reliability of shared data is also guaranteed. We conducted some simulation experiments and the experimental results showed that the proposed architecture could maintain good training performances when dealing with a small number of malicious nodes and differentiated data quality, which has good fault tolerance, and can be applied to edge computing scenarios.

## 1. Introduction

The machine learning method is based on sample data training to obtain machine learning models suitable for different tasks and scenarios. These sample data are generally collected from different users, terminals, and systems and stored centrally. In practical application scenarios, this way of collecting sample data faces many problems. On one hand, this approach compromises the privacy and security of the data. In some applications such as Internet of Things devices, which are limited by the requirements of data privacy and security, it is impossible to realize the centralized storage of data. On the other hand, this method will increase the communication overhead. In some applications that rely heavily on mobile terminals such as the Internet of Things, the communication overhead cost of this data aggregation is enormous.

Federated learning allows for multiple users (or clients) to collaborate on training a shared global model without sharing data on the local device. Multiple rounds of federation learning are coordinated by a central server to obtain the final global model [1]. At the beginning of each round, the central server sends the current global model to the clients involved in federated learning. Each client trains the received global model based on its local data and returns the updated model to the central server when it has finished training. The central server collects the updates returned by all the clients and performs a single update to the global model, thus ending the round.

However, federation learning for IoT networks faces a number of challenges in its implementation. First, existing federation learning frameworks are based on a centralized model coordinator, which still face serious security challenges such as a single point of failure. Second, the gradient data learned by nodes in federation learning indirectly reflect the information of the training samples, and an attacker can invert the sample data from valid gradient information, requiring a reduction in gradient communication to reduce the possibility of privacy compromise.

Permissioned blockchain is blockchain managed by a number of institutions, each of which operates one or more nodes, and only the permitted nodes can participate in voting, bookkeeping, and block building [2]. Each node in the blockchain usually has a physical body or organization that corresponds to it. Participants are authorized to join the network and form stakeholder permissions to work together to maintain the blockchain. The data in the chain can only be read, written, and sent by different institutions within the system, and the data are recorded jointly. It has the advantages of relatively fast transactions, no mining, low transaction costs, fast transactions, and support for regulatory interests.

Through the authorization mechanism and identity management of the permissioned blockchain, mutually untrustworthy users can be integrated together as participants to create a secure and trustworthy cooperation mechanism. The model parameters of federated learning can be stored in the permissioned blockchain, ensuring the security and reliability of the model parameters. For example, while multiple participants perform federation learning, the permissioned blockchain is deployed to record the data fingerprint (including modeling samples, inference samples, and interaction information) of the federation learning, while the corresponding raw data are stored locally by the participants. When a malicious attack is detected on a sample, an investigation team is formed by each participant or a third party to verify the original data based on the fingerprints recorded in the permissioned blockchains to find out which party has been attacked and take appropriate measures. Through the distributed ledger feature of the permissioned blockchains, it is naturally guaranteed that the model parameter data consistency, synchronization, and sharing between multiple participants in the federation learning are secure and trustworthy, and that the model parameter data interaction is transparent, traceable, tamper-proof, and anti-forgery.

To address these challenges, we propose an asynchronous federal learning system based on permissioned blockchains that addresses the federated learning single point of failure problem and data security privacy issues. The major contributions of this paper are summarized as follows:(1)A permissioned blockchain-based federated learning framework is proposed. The permissioned blockchains are composed of a main-blockchain and multiple sub-blockchains, each of which is responsible for partial model parameter updates and the main- blockchain is responsible for global model parameter updates.(2)A multi-chain asynchronous model aggregation algorithm is proposed, which uses deep reinforcement learning for node selection, the sub-blockchain nodes audit the gradient and proof of correctness of the encryption and partially aggregate the model parameters, and the main blockchain is responsible for the global model parameter updates.(3)A prototype permissioned blockchain-based federated learning system was implemented and extensive experiments were conducted to demonstrate its feasibility and effectiveness.

The rest of the paper is organized as follows. In Section 2, we explain the concepts of blockchain, federation learning, and reinforcement learning used in this paper. Section 3 presents related work. We introduce our design of the system in Section 4. Section 5 presents the experimental simulation results showing that our technique is working. Finally, we conclude the paper and identify future directions in Section 6.

## 2. Background

### 2.1. Federated Learning

Federated learning is an emerging fundamental AI technology first proposed by Google in 2016 to solve the problem of updating models locally for Android phone end-users [3]. Efficient machine learning is carried out between multiple participants or computational nodes. The machine learning algorithms that can be used for federated learning are not limited to neural networks, but also include important algorithms such as random forests. Federated learning is expected to be the basis for the next generation of collaborative algorithms and collaborative networks for artificial intelligence.

Features of federated learning: data from all parties are kept local, without compromising privacy or violating regulations; multiple participants combine data to build a fictional shared model and benefit from the system together; each participant has the same identity and status under the federated learning system; the modeling effect of federated learning is the same as, or not significantly different from, modeling the entire dataset in one place (under user alignment or feature alignment of the individual data or feature alignment). Transfer learning can be used to achieve knowledge migration by exchanging cryptographic parameters between data, even if the users or features are not aligned. Federated learning allows two or more data-using entities to collaborate and use data together without leaving the local area, solving the problem of data silos. A typical federal learning system is shown in Figure 1.

The existing federal learning method based on homomorphic encryption is generally as follows: each device side trains a local model using a local dataset, obtains the local model gradient information after training, encrypts the local model gradient information after training using a homomorphic encryption algorithm (such as the Paillier algorithm, etc.), and sends the local model gradient information after encryption to the server-side. Then, after receiving the respective encrypted local model gradient information from each device, the server-side aggregates the encrypted local model gradient information according to a predetermined aggregation method to obtain the global model gradient information in the encrypted state (global model gradient information in the form of cipher text). The server sends the global model gradient information in the encrypted state to each device, so that each device decrypts the global model gradient information in the encrypted state received by each device using the homomorphic encryption algorithm, and continues to train the local model based on the decrypted global model gradient information using the local dataset until the local model converges or reaches the number of iterations of training to obtain the global model. However, although this processing provides strong privacy guarantees for federation learning using homomorphic encryption, it performs complex encryption operations (e.g., modulo multiplication or exponential operations, etc.), and this complex encryption operation is very time-consuming and requires a lot of computational resources. At the same time, the complex encryption operation results in a larger ciphertext, which consumes more network resources during transmission than during plaintext transmission.

### 2.2. Permissioned Blockchains

The blockchain was introduced in 2008 by a person named Satoshi Nakamoto as Bitcoin [4]. Bitcoin blockchain individuals are connected through a peer-to-peer network in order to publish financial transactions based on encryption using public and private keys. A blockchain block is a basic component containing a block header and a block body, as shown in Figure 2. These two functions contain multiple pieces of information such as header number, nonce, current hash, previous hash, signed transaction data, etc.

Blockchain platforms can be divided into two categories: public blockchains and permissioned blockchains. In a public blockchain, all nodes can participate in voting, bookkeeping, and block building. Permissioned blockchains are blockchains jointly managed by several organizations, each of which runs one or more nodes, and only permitted nodes can participate in voting, bookkeeping, and block building. Permissioned blockchain has the characteristics of fast transaction speed, no mining, low transaction cost, and support for regulation. According to the application requirements, permissioned blockchain supports various consensus protocols such as Byzantine Fault Tolerance, RAFT, Practical Byzantine Fault Tolerance, Plenum, etc. Each node on the blockchain usually has an entity or organization corresponding to it, and the participants are authorized to join the network and form a stakeholder permit to jointly maintain the blockchain operation. The data in it only allow different institutions in the system to read, write, and send transactions, and work together to record transaction data.

The advantages of the permissioned blockchains are as follows:(1)Strong controllability. Compared with public blockchains, public blockchains generally have many nodes, and once a blockchain is formed, the block data cannot be modified. For example, Bitcoin has many nodes, and it is impossible to change the data in it if you want to modify. In contrast, in permissioned blockchains, data can be modified as long as the majority of pre-selected nodes reach consensus.(2)Better performance. The permissioned blockchain is to some extent owned only by the members within the permit as the number of nodes in the permit is limited, so it is easy to reach a consensus.(3)Fast transaction speed. Only permissioned nodes can join the blockchain network, and transactions can only be verified by consensus nodes without network-wide confirmation. In a way, the essence of permission is still a private blockchain, it has a limited number of nodes, and it is easy to reach consensus, so the transaction speed is also relatively blocky.(4)Better privacy protection. The user identity is managed and the read access is restricted, which can provide better privacy protection. The data of the public blockchain is public, but the permission is different, so only the permission internal organization and its users have the permission to access the data.

Better privacy protection can be provided by managing the identity of the user with restricted read access. Data in the public chain are public, but licensing is different in that only licensed internal agencies and their users have access to the data.

The permissioned blockchain system is generally divided into a storage layer, a blockchain core layer, a blockchain service layer, an interface layer, and an application layer, as shown in Figure 3. The storage layer is responsible for various cached data storage and persistent storage of blockchain data. The blockchain core layer is responsible for core blockchain functions such as consensus mechanism, reputation mechanism, user data, transaction data, smart contracts, encryption and decryption, signature verification, authentication management, and node management.

### 2.3. Reinforcement Learning

Reinforcement learning, a subfield of machine learning, is inspired by the behaviorism theory in the psychology of how intelligence gradually develop expectations of stimuli in response to rewards or punishments given by the environment, producing habitual behavior that maximizes benefits. It emphasizes how to act based on the environment to maximize the expected benefit.

Reinforcement learning is the process by which an agent learns by ‘trial and error’ and is guided by the rewards it receives from interacting with the environment to maximize the rewards it receives [5]. The signal is an evaluation of how well the action was produced (usually a scalar signal), rather than telling the reinforcement learning system (RLS) how to produce the correct action. Since the external environment provides little information, the RLS must learn from its own experience. In this way, the RLS acquires knowledge in an action-evaluation environment and improves the course of action to suit the environment.

Figure 4 illustrates the principle of reinforcement learning. The Agent selects an action to be used in the environment, and the environment accepts the action with a state change, and at the same time generates a reinforcement signal (reward or punishment) back to the Agent. The choice of action affects not only the immediate reinforcement value but also the state of the environment at the next moment and the final reinforcement value. The goal of Agent is to find the optimal strategy in each discrete state to maximize the expected discount and reward.

## 3. Related Works

In this section, we review the most recent related works in the field of asynchronous federated learning (see Section 3.1) and in the area of blockchain-based asynchronous federated learning (see Section 3.1).

### 3.1. Asynchronous Federal Learning

The purpose of research on asynchronous federation learning is to give more flexibility to participating federation learning nodes. Wang Hao et al. [6] proposed the model fusion method, which can also be understood as decision fusion, and is averaged over the prediction results of multiple network models to improve learning accuracy and has been widely used (e.g., to improve ImageNet recognition performance) [7,8,9,10]. McMahan Brendan et al. [3] presented an asynchronous learning strategy that presents an aggregation of temporally weighted local models on the server. The time-weighted aggregation strategy is introduced on the server to exploit the previously trained local models, thus improving the accuracy and convergence of the central model. Chen Zunming et al. [11] proposed a lightweight dynamic asynchronous algorithm that considers averaging frequency control and parameter selection for federal learning to speed up model averaging and improve efficiency. Wang Qizhao et al. [12] introduced an efficient asynchronous federation learning that allows edge nodes to select some models from the cloud for asynchronous updates based on local data distribution, thus reducing computation and communication and improving the efficiency of federation learning. Li Ming et al. [13] proposed a novel asynchronous vertical federation learning framework with gradient prediction and double-ended sparse compression to accelerate the training process and reduce the transmission of intermediate results. Chen Zheyi et al. [14] developed an asynchronous federation learning scheme that employs a lightweight node selection algorithm to efficiently perform the learning task by iteratively selecting heterogeneous IoT nodes to participate in global learning aggregation. Yang Helin et al. [15] developed an asynchronous federated learning (AFL) framework that could provide asynchronous distributed computing by implementing model training locally without transmitting raw sensitive data to the UAV server. Asynchronous dominant actor criticism (A3C) based federated device selection, drone placement, and resource management algorithms were used to improve the speed and accuracy of federated convergence. Zhang Hongyi et al. [16] developed a real-time end-to-end federation learning framework using sliding training windows to reduce communication overhead and speed up model training. Sun Wen et al. [17] proposed a clustering-based asynchronous federation learning framework using Lyapunov dynamic missing queues and deep reinforcement learning (DRL) to adaptively adjust the aggregation frequency of federation learning to improve the learning performance under resource constraints. Xue M. A. et al. [18] proposed a collaborative federation learning mechanism to construct a hierarchical multi-level confidential communication network with the asynchronous fusion of network parameters uploaded by the respective fusion centers using sequential Kalman filtering algorithms at the cloud and edge, respectively. Chen Yujing et al. [19] proposed an asynchronous online federated learning (ASO-Fed) framework in which edge devices perform online learning with continuous local data streams and a central server aggregates model parameters from clients. Xiaofeng et al. [20] proposed a privacy-preserving asynchronous federation learning mechanism (PAFLM) for edge network computing, which allows multiple edge nodes to achieve more efficient federation learning without sharing their private data. Lu Yunlong et al. [21] proposed a different private asynchronous federation learning scheme for resource sharing in vehicular networks that incorporates local difference privacy into federation learning to protect the privacy of updated local models with a stochastic distributed update scheme. Ma Qianpiao et al. [22] proposed a semi-asynchronous federation learning mechanism (FedSA) where the parameter server aggregates a certain number of local models in each round in the order of arrival. Liu Jianchun et al. [23] presented a new communication-efficient asynchronous federated learning (CE-AFL) mechanism in which the parameter server will only aggregate local model updates from a certain fraction α (0 < α < 1) of all edge nodes in the order of their arrival in each epoch. Rizk Elsa et al. [24] proposed dynamic federated learning (DFL) in which a random subset of available agents is updated locally based on their data in each iteration. Miao Qinyang et al. [25] proposed a hierarchical asynchronous federated learning (FL) framework based on sensitive task decomposition using deep reinforcement learning (DRL) techniques to select participants with sufficient computational power and high-quality datasets. By integrating task decomposition and participant selection, reliable data sharing is achieved by sharing local data models instead of source data, while protecting data privacy. Agrawal Shaashwat et al. [26] presented a novel temporal model averaging algorithm, which uses a dynamic expectation function to calculate the number of expected client models per round and a weighted averaging algorithm to successively modify the global model. Gao Yujia et al. [27] designed a resilient local update algorithm that can train personalized models by setting specific update weights for each node based on the differences between the global and local models. Wang Zhongyu et al. [28] proposed a new asynchronous FL framework that takes into account the potential failures of uploading local models and the resulting lag in global updates of models of different magnitudes. Liu Kai-Hsiang et al. [29] proposed an online solution based on actor-critical federation learning called AC-Federate, where an actor-critic (AC) model for each edge node jointly optimizes continuous actions (i.e., radio and computational resource allocation) and discrete actions (i.e., offloading decisions) and trains the model using a weighted loss function. Each edge node uploads the gradients of its actor-critic neural network to the central controller in an asynchronous manner, updating all edge nodes with the integrated network parameters.

### 3.2. Blockchain-Based Federated Learning

Lu et al. [30] presented a secure data sharing architecture authorized by the blockchain, which stores and shares the federal learning model and parameter transfer process through the blockchain to ensure the security of the sharing process. Kang et al. [31] proposed a scheme based on reputation as a reliable metric to select trusted workers for federated learning that used a multi-weight subjective logic model to design an efficient reputation computation scheme based on the interaction history of task publishers and recommended reputation records, and manage reputation using a blockchain running on edge nodes. Qu Y. et al. [32] and Qi Y. et al. [33] applied local differential privacy techniques to blockchain federation learning to protect data privacy in the industrial Internet and smart transportation domains by adding noisy perturbations to the raw data. Liu et al. [34] used smart contracts containing validation datasets to automatically evaluate updates uploaded by devices before performing model aggregation to detect the presence of poisoning attacks. Kim et al. [35] put forward a federated learning method applied to the device side based on the blockchain framework where the local gradients of each iteration are stored in blocks after verification and consensus, and analyzed the end-to-end latency and optimal block generation rate. Wang et al. [36] proposed a blockchain federated learning system supporting heterogeneous models and designed two mining methods offline and online to resist Byzantine. Y. Lu et al. [37] presented a hybrid blockchain architecture based federal learning architecture that consists of a permissioned blockchain and a local directed acyclic graph (DAG) using deep reinforcement learning (DRL) for node selection, integrating the learned model into the blockchain and performing two-stage validation. By combining blockchain with federation learning, Y. Lu et al. [37] endowed the system with the optimized decision-making capability of artificial intelligence while retaining the security and trustworthiness of the blockchain, and it also optimized the blockchain in terms of efficiency and communication to enhance its operation. Kang et al. [36] proposed a scheme to select trusted workers for federal learning based on reputation as a reliable metric using a multi-weighted subjective logic model based on the task publisher’s interaction history and recommended reputation records, designed an efficient reputation calculation scheme, and managed the reputation using a blockchain running on edge nodes. Y. J. Kim et al. [38] considered two types of weight selection for a blockchain-based federated learning scenario to update a subset of clients of the global model, which possesses relatively high stability and can improve the convergence speed of the federated learning task model.

Liu Yinghui et al. [39] considered federated learning with stagnation coefficients (FedAC) while using a blockchain network instead of a classical central server to aggregate global models. It avoids real-world problems such as anomalous local device training failure interruptions, dedicated attacks, etc. Lu Yunlong et al. [40] presented an asynchronous federation learning scheme to improve efficiency by employing deep reinforcement learning (DRL) for node selection. The reliability of the shared data is also ensured by integrating the learned model into the blockchain and performing a two-stage verification. Lu Y. et al. [40] suggested a blockchain-empowered federation learning scheme to enhance communication security and data privacy protection in digital twin edge networks (DITEN) using digital twin-empowered reinforcement learning to schedule relay users and allocate spectrum resources. Liang Hao et al. [41] put forward a framework for smart driving model sharing and collaborative training among operators using blockchain technology. In this framework, smart contracts are used to enable the management of reinforcement learning across the federation. Liu Wei et al. [42] proposed a blockchain-based health care data sharing scheme that used blockchain as an incentive mechanism for rewarding health care providers who are honest with high-quality data or contribute to decryption.

## 4. Asynchronous Federal Learning System Based on Permissioned Blockchains

### 4.1. System Overview

The asynchronous federated learning system based on permissioned blockchains is divided into four layers of architecture: the IoT device layer, the network layer, the edge computing layer, the blockchain layer, and the application layer. Figure 5 shows an overview of our system architecture.

(1)IOT device layer

IOT device communication is carried out through 4G, 5G, NB-IoT, Ethernet, serial bus, parallel bus, and other common physics. The communication protocol can support http, mqtt, Canbus, Modbus, CC-link, etc. Devices include QR code tags and readers, RFID tags and readers/writers, cameras, GPS, various sensors, video cameras, terminals, sensor networks, and other data collection devices. RFID technology, sensing and control technology, and short-range wireless communication technology are the main technologies involved in the sensing layer. The sensing layer consists of intelligent sensing nodes and data collection nodes. Smart nodes sense various kinds of information, for example, Smart Dust, which is used for environmental information collection, can sense temperature, humidity, graphics, and other information. These smart nodes can form a network and network themselves to pass data to an upper layer gateway access point, which submits the collected sensing information through the network layer to the backend for processing. Applications such as environmental monitoring and pollution monitoring are based on this type of architecture for the IoT.

(2)Edge computing layer

Store and process data at the edge devices without the need for a network connection for cloud computing. This eliminates the need for high bandwidth constant network connections. The edge computing layer includes edge data cleansing, edge data storage, alarm message pushing, and edge device response. Edge device data cleansing: Edge computing devices perform simple cleansing of data based on their own computing power such as checking data consistency, handling invalid and missing values, etc. Edge data storage: The edge computing node stores data according to the storage capacity of the device and uploads data as the network allows. Alarm information push: Alarm information of edge computing node devices is pushed to relevant personnel by SMS, email, etc. Edge device response: The edge computing borrowing point makes device response according to the characteristics of the edge device to ensure the real-time response of the device and the consistency of the device performance requirements.

(3)Blockchain layer

The blockchain layer is the core layer of the IoT combined with blockchain, and the most important thing in the blockchain layer is the consensus algorithm. The blockchain system is generally divided into a storage layer, a blockchain core layer, a blockchain service layer, an interface layer, and an application layer, as shown in Figure 1. The storage layer is responsible for various cached data storage and persistent storage of blockchain data. The blockchain core layer is responsible for blockchain core functions such as consensus mechanism, reputation mechanism, user data chain, transaction chain, smart contract, encryption and decryption, signature verification, authentication management, and node management. The blockchain consensus algorithm is to allow blockchain nodes to reach consensus on the creation, verification, and storage of blocks and to ensure the consistency of blockchain copies in the system service layer. The smart contract layer includes two major parts: contract management and contract interface. Among them, contract management is responsible for the deployment, installation, debugging, and operation of smart contracts. The contract interface is provided to external systems for invocation.

(4)Application layer

The application layer consists of various application servers (including database servers), whose main functions include the aggregation, conversion, and analysis of collected data as well as the adaptation and event triggering of user-level presentation. A large amount of raw data obtained from the end nodes is transformed into data of practical value after transmission, conversion, and analysis at the network layer; the application servers, which store these data, will adapt the information presented according to the user’s presentation device and trigger relevant notification messages according to the user’s settings. At the same time, when it is necessary to complete the control of the end nodes, the application layer can also complete the control command generation and command distribution control.

The application layer shall provide users with the user interface (UI interface) of the Internet of Things application including user equipment (such as PC, mobile phone), client, etc., in addition, the application layer also includes a cloud computing function. Based on cloud computing, the Internet of Things Management Center, information center, and other departments can intelligently process massive information.

### 4.2. Asynchronous Federated Learning Algorithm

To improve data security, training efficiency, and accuracy, we designed an asynchronous federated learning algorithm based on permissioned blockchains for our federated learning scheme, as shown in Figure 6. The permissioned blockchains are composed of a main-blockchain and multiple sub- blockchains that are responsible for synchronous global aggregation and asynchronous local training, respectively, in our asynchronous federated learning scheme. 

The asynchronous federated learning algorithm proposed in this paper consists of three phases: node selection, local training, and global aggregation. Node selection is performed by using reinforcement learning algorithms to select participating blockchain nodes to formulate and solve an optimization problem. The permission chain acts as a centralized server for collecting federated learning models. The permission chain is composed of a main-blockchain and multiple sub-blockchains, each responsible for partial model parameter updates and the main-blockchain responsible for global model parameter updates. Specifically, we will follow the following seven detailed steps for training.

Step 1: Terminal devices and edge computing devices are registered and authenticated, each terminal device and edge computing device is registered with the sub blockchain, and the sub-blockchain authenticates each device and issues certificates.

Step 2: The global model is assigned to the main-blockchain, the sub-blockchain downloads the global model from the main-blockchain, and the participating edge computing nodes are selected based on the nodes for deep reinforcement learning, considering factors such as device training latency, model transmission latency, and accuracy rate.

Step 3: The participating edge computing node downloads the global model from the sub-blockchain and initializes the training model and parameters.

Step 4: The end device loads data samples and offloads them to the edge computing device for local model training. We use the convolutional neural network (CNN) layer as a feature extractor to extract features from the raw data in the mobile device, and after n iterations, the resulting gradient, giving the encrypted gradient and proof of correctness, is encrypted and uploaded to the sub-blockchain.

Each node adjusts the gradient based on the received gradient clipping criteria, as shown in Equation (1).
(1)$${\mathrm{Adj}}_{c}\left(g_{k}\left(w_{t},\xi_{i}\right)\right)=\{\begin{array}{c} g_{k}\left(w_{t},\xi_{i}\right),\\ \frac{cg_{k}\left(w_{t},\xi_{i}\right)}{∥g_{k}\left(w_{t},\xi_{i}\right)∥}, \end{array}\begin{array}{c} ∥g_{k}\left(w_{t},\xi_{i}\right)∥\leq c\\ ∥g_{k}\left(w_{t},\xi_{i}\right)∥\geq c \end{array}$$

Each node is trained locally and the gradients are encrypted as shown in Equation (2).
(2)$${\tilde{g}}_{k}\left(t\right)=\mathrm{encrypt}({\mathrm{Adj}}_{c}\left(g_{k}\left(w_{t},\xi_{i}\right)\right)),\forall i\in {ℬ}_{k,t}\;$$

Step 5: The sub-blockchain nodes review the gradient and proof of correctness of the encryption. Collaborative decryption requires at least 2/3 participants to provide their private share to decrypt a cipher as a way to further enhance the privacy of the data. The sub-blockchain performs a local aggregation of the model parameters to aggregate the model parameters. The resulting gradient, which gives the encrypted gradient and proof of correctness, is encrypted and then uploaded to the main-blockchain.

The sub-blockchain receives the encryption gradient and evidence from the node for verification, and after the verification passes, decrypts the gradient and uploads it to the main-blockchain, as shown in Equations (3) and (4).
(3)$$g_{nk}\left(t\right)=decrypt\left({\tilde{g}}_{k}\left(t\right)\right)$$
(4)$${\bar{g}}_{k}\left(w_{t}\right)=\frac{1}{\left|B_{k,t}\right|}\sum_{i\in {ℬ}_{k,t}}\left({\mathrm{Adj}}_{c}\left(g_{nk}\left(w_{t},\xi_{i}\right)\right)+\eta_{i}\right)\;$$

Step 6: The main-blockchain node reviews the gradient and proof of correctness of the sub-blockchain encryption and performs global aggregation of the model parameters to aggregate the model parameters. We improved the aggregation efficiency by dividing the aggregation phase into a local aggregation phase and a global aggregation phase. For each node, local aggregation is performed asynchronously between nodes within a local range to improve the quality of the trained local model. The main blockchain judges whether the preset convergence conditions of the model are met. If not, it will carry out the next round of training. If it arrives, it will terminate the federal learning task.

The main blockchain updates the global model covariates using the average of the *K* gradients, as shown in Equation (5).
(5)$$w_{t+1}=w_{t}-\gamma \frac{1}{K}\sum_{i=1}^{K}{\bar{g}}_{k}\left(w_{t-\tau \left(t\right)}\right)$$

Step 7: Repeat steps 2–6 until the model converges or reaches a predetermined number of training rounds.

The permissioned blockchain uses all gradients to update the parameters of the collaborative model that all participants have collaboratively encrypted, and the updated model parameters are obtained by collaborative decryption. To prevent malicious participants from providing incorrect gradients or giving incorrect decryption shares during the decryption phase, participants are required to provide encrypted gradients and proof of correctness before uploading the gradients, and to allow third parties to audit the participants’ validation as a means of ensuring the auditability of the data. After the parameters have been updated, participants need to provide their decryption shares and the corresponding proof of correctness to download the co-decryption parameters, and again, any third party can audit the decryption shares for correctness. Algorithm 1 provides the complete process of our proposed blockchain-based asynchronous federation learning scheme.
**Algorithm 1****:** Blockchain-Based Asynchronous Federated Learning Algorithms.**Input:** Initial network status and blockchain nodes. Initial global model θ^g^ and local models θ^l^. The registering edge nodes as participating nodes EI = {e1, e2, …, eN}. The dataset d_i_∈D.**Input:** Select the participating edge nodes E_p_ ⊂ E_I_ by running node selection algorithm.1:  **for** episode∈{1,EP} **do**2:   **for** s_episode∈{1,EP_s_} **do**3:    sub-blockchain retrieves global model from main-blockchain4:    **for** each edge node e_i_ ∈E_I_
**do**5:     e_i_ executes the local training on its local data d_i_, according to Equations (1)–(3), encrypted gradient and proof of correctness, upload to sub-blockchain.6:     According to Equations (3) and (4), the sub-blockchain node reviews the encrypted gradient and proof of correctness, performs local aggregation asynchronously, update local models, gives the encrypted gradient and proof of correctness, and after encryption, uploads it to the main-blockchain.7:     According to Equation (5), the main-blockchain node reviews the encrypted gradient and global aggregation of model parameters.8:   **end for**9:  **end for**10:  Repeat steps 2–6 until the model converges or reaches a predetermined number of training rounds.11:  **end for**12: **return** The parameters of the final global model parameters.

### 4.3. Node Selection Algorithm

The choice of device node is influenced by a number of factors. First, the differential computing and communication capabilities of the end devices directly affect local training and data transfer latency. Second, the size of the dataset carried on the end devices varies and the data may not satisfy the independent homogeneous distribution property, which makes a difference in the training quality of the local model. The number of end devices involved in federated learning training in edge networks is often large, and when dealing with the node selection problem, the traditional actor-critic algorithm is difficult to determine the learning rate, which may lead to slow convergence or premature convergence and other disadvantages, and the convergence performance of the algorithm needs to be improved. Therefore, this paper designed a DPPO-based node selection algorithm based on the idea of PPO algorithm design. The detailed process is shown in Algorithm 2.

After executing a step according to a certain end-device node selection policy, the environment data changes and a reward value for evaluating this behavior is obtained, and the reward function is as in Equation (6).
(6)$$r_{t}=\frac{1}{\sum_{d\in D}\beta_{d}}\underset{s\in S_{i}}{\sum }\underset{d\in D}{\sum }A_{i}\beta_{d}$$

In Equation (6), $A_{i}$ is the loss function of the test set and $\beta_{d}$ is whether the node is selected or not.

The selection policy π is shown in Equation (7).
(7)$$\alpha_{i,t}=\pi \left(s_{t}\right)$$

After the corresponding state adopts the corresponding action according to this policy to maximize the expectation of the goal-cumulative return of reinforcement learning, the optimization policy is shown in Equation (7).
(8)$$\pi^{*}=\underset{\pi }{\arg max}E\left[\underset{t=0}{\sum }\sigma^{t}\cdot r_{t}\right]$$

In Equation (8), $\sigma^{t}$ is the discount factor and $r_{t}$ is the reward value.

The optimization goal of the network model is to satisfy Equation (9) by updating the parameter *θ* as shown.
(9)$$\underset{\theta }{max}E\left[\frac{\pi_{\theta \left(\alpha |s\right)}}{\pi_{\theta_{old}\left(\alpha |s\right)}}Adv_{\theta_{old}}\left(s,\alpha \right)\right]$$

In Equation (9), $Adv_{\theta_{old}}\left(s,\alpha \right)$ is the dominance function and $\pi_{\theta \left(\alpha |s\right)}$ is the probability of taking action π in state s based on the end-device node selection policy α.
(10)$$Adv_{t}=\underset{i>t}{\sum }\sigma^{i-t}r_{i}-V_{\varphi }\left(s_{t}\right)$$

In Equation (10), σ is the discount factor; V is the state value function; and φ is the critic network parameter.
(11)$$L\left(\phi \right)=-\underset{t=1}{\sum }{\left(\underset{i>t}{\sum }\sigma^{i-t}r_{i}-V_{\varphi }\left(s_{t}\right)\right)}^{2}$$

The loss function of the critic network is calculated using Equation (11) and back propagated to update the critic network parameter φ.
**Algorithm 2:** The Node Selection Algorithm Based on DDPO (BAFL-DPPO).**Input:** Initial network status and task information.**Input:** Initialization of network, equipment and task information and global network parameters.1:  **for** episode∈{1,EP} **do**2:   **for** s_episode∈{1,EP_s_} **do**3:     $\mathrm{Each}\;\mathrm{node}\;\mathrm{according}\;\mathrm{to}\;\mathrm{the}\;\mathrm{global}\;\mathrm{PPO}\;\mathrm{policy}\;\alpha_{i,t}=\pi \left(s_{t}\right)\;\mathrm{execute}\;\mathrm{node}\;\mathrm{selection}\;\mathrm{action}\;\alpha$.4:    Calculate the reward rt according to Equation (6), select the next state *st*+1 according to Equation (8), and store the current state, action and reward as samples.5:    Update current network and device status information.6:    **end for**7:    Each node uploads the collected data synchronously to the global network services.8:    Update dominance function and actor1 network parameters *θ*.9:    Back propagation update critic network parameters *ϕ*.10:    **if** s_episode%circke == 0 **do**11:     Update actor2 with the parameters in actor112:    **end if**13: **end for**14: **return** Selected node list.

## 5. Simulation Experiments

### 5.1. Experimental Configuration

In order to simulate a real-world scenario, we set up the following experimental environment: a GPU server with high computational power acts as the parameter server and is responsible for most of the computational work; the remaining computers simulate individual learning nodes in the edge network, each performing federal learning independently with a bandwidth of 1 Mbps. Each computer stores some of the data locally and trains the neural network model based on the local data alone. The experiments used our self-developed federated blockchain system, consisting of a main blockchain and three sub-blockchains, using the HotStuff consensus algorithm [43] on each blockchain. In this paper, we studied multiple edge device nodes with different computational power, and simulated these devices with different computational power by adding a pause interval. To better simulate the experimental scenario, the federal learning on these computers was controlled by a separate computer.

We evaluated the proposed asynchronous federation learning on the MNIST dataset, which is partitioned into 100 slices that are assigned to 100 providers. The edge data sharing task is to share the computational results on the local data of each data provider. We used a convolutional neural network (CNN) model as the local training model. In each iteration, there is one global aggregation and 10 local trainings. In addition, we used a local CNN model and a centralized CNN model as benchmark algorithms on the same dataset. The local CNN trains the model on the local provider’s dataset, while the centralized CNN model is trained on the entire centralized dataset. We then validated the performance of the DDPO-based node selection algorithm.

The DPPO algorithm uses seven threads to interact with the external environment and the reward discount factor was set to 0.9. The learning rates of the actor and critic networks were set to 0.0001 and 0.0002, respectively, and actor 2 was updated using the parameters in actor, whenever the agent was trained for 100 rounds. Two algorithms were selected as the comparison of the proposed algorithm (BAFL-DPPO) in this paper. (1) BAFL-Greedy: This algorithm selects all device nodes for model aggregation in each iteration of federal learning training. (2) Local Training: No federal learning mechanism is used, and model training is performed only on local devices.

### 5.2. Experimental Results

First, we evaluated the accuracy and loss of the proposed solution for different numbers of data providers on the MNIST dataset, which is a classification problem, so the accuracy in the experiment can be defined as the number of correct classifications as a percentage of the total number of samples. The variation of the accuracy of the three algorithms with 30% of malicious device nodes is presented in Figure 7, from which it can be seen that the accuracy of the models obtained by the three algorithmic mechanisms is low in the early stages of training, which indicates that the training accuracy of the models needs to be guaranteed by a sufficient number of training iterations. When the number of iterations reached 10, the accuracy of the models obtained by the three mechanisms stabilized, with the accuracy of BAFL-DPPO, BAFL-Greedy, and Local Training stabilizing around 0.82, 0.75, and 0.7, respectively. The BAFL-DPPO algorithm maintains good training performance with a small number of malicious nodes and varying data quality, while Local Training has difficulty in ensuring the training quality. The gain of the BAFL-DPPO algorithm is more significant in the case of data heterogeneity due to the reinforcement learning based dynamic adjustment strategy.

The variation of the loss functions of the three algorithms with 30% of malicious device nodes is presented in Figure 8, from which it can be seen that the loss function of Local Training always fails to converge and is significantly higher than that of BAFL-DPPO and BAFL-Greedy because the federal learning mechanism is not used. Greedy loss rate stabilized when the number of iterations reached 10, and BAFL-DPPO converged faster than BAFL-Greedy with the smallest value of the loss function.

The latency comparison results of the three algorithms are shown in Figure 9, from which it can be seen that the latency of the algorithm did not increase or decrease directly with the increase in nodes, but tended to be stable at about 8 s. The latency of the BAFL-Greedy algorithm increased with the increase in nodes, but was lower than that of the Local Training algorithm. The BAFL-DPPO algorithm ensures low latency when dealing with a wide range of node numbers due to its ability to efficiently select device nodes with high training quality for model aggregation.

To test the failure probability of malicious attacks, we randomly selected a certain percentage of nodes as malicious nodes. Figure 10 illustrates the comparison of the failure probability of the three methods in the face of malicious attacks: the failure probability of the Local Training algorithm increased with the increase in malicious device nodes. The BAFL-Greedy and BAFL-DPPO algorithms were Byzantine fault-tolerant when the percentage of malicious device nodes was less than 33%, and the failure probability increased with the increase in malicious device nodes when the percentage of malicious device nodes exceeded 33% attack capability.

## 6. Conclusions

In this paper, we first propose an asynchronous federated learning system based on permissioned blockchains that can effectively alleviate the overhead of the synchronous federated learning algorithm on the synchronization problem, while the reliability of shared data is guaranteed to improve the learning performance of federated learning. According to our simulation experimental results, we confirm the effectiveness of our proposed scheme in terms of efficiency and accuracy with good convergence and robustness, enough to maintain a good training performance even when dealing with a small number of malicious nodes and differential data quality, providing an effective solution for performing federated learning at network edge devices.

## Figures and Tables

**Figure 1 sensors-22-01672-f001:**
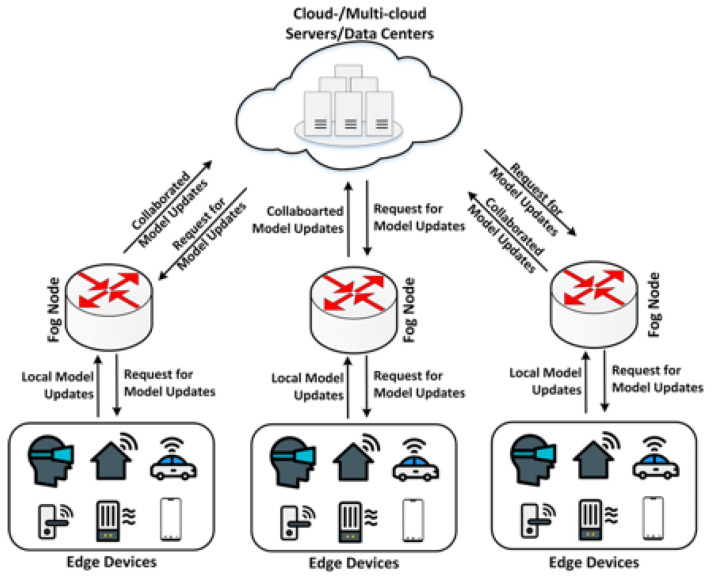
Federal learning system.

**Figure 2 sensors-22-01672-f002:**
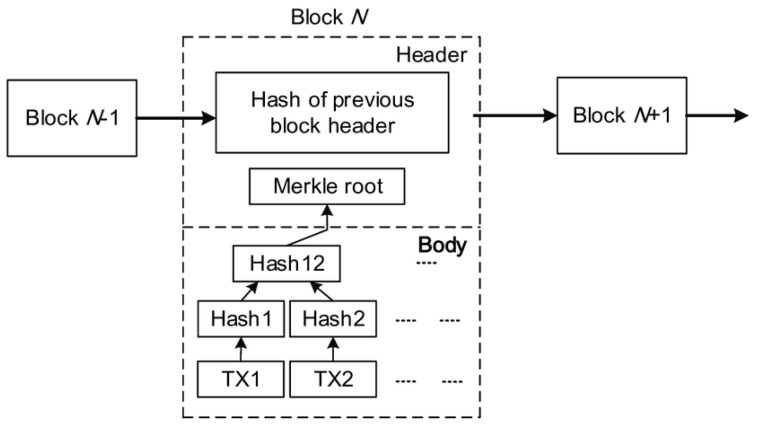
Blockchain structure.

**Figure 3 sensors-22-01672-f003:**
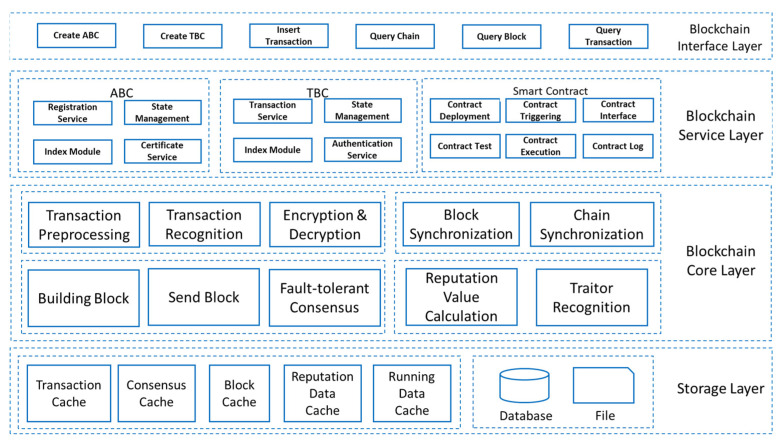
Permissioned blockchain system architecture.

**Figure 4 sensors-22-01672-f004:**
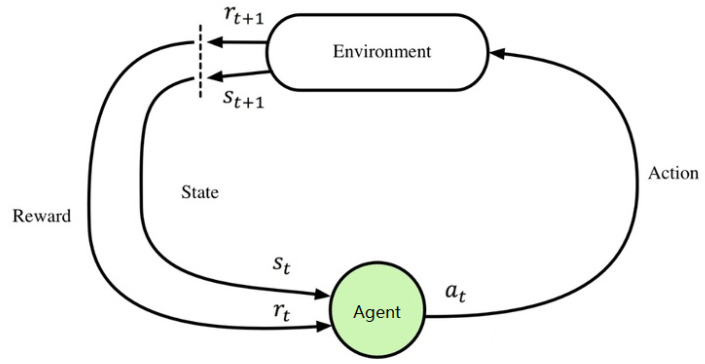
Schematic diagram of the principle of reinforcement learning.

**Figure 5 sensors-22-01672-f005:**
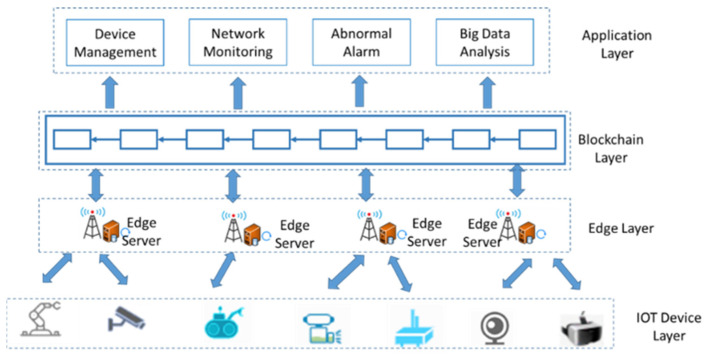
Asynchronous federal learning system based on permissioned blockchain.

**Figure 6 sensors-22-01672-f006:**
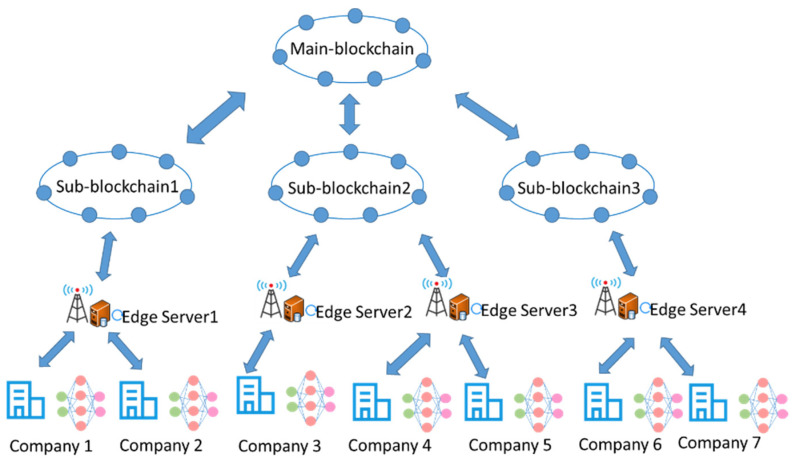
Schematic diagram of the asynchronous federation learning algorithm.

**Figure 7 sensors-22-01672-f007:**
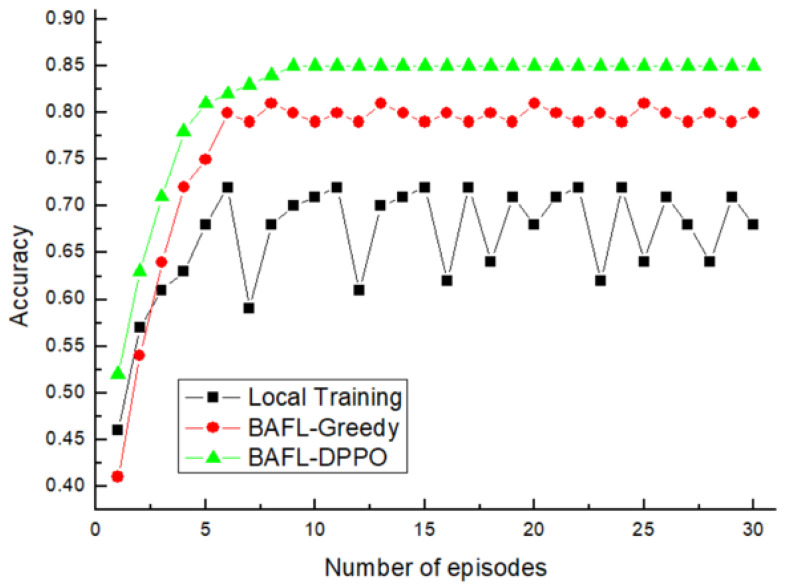
Accuracy comparison (30% of malicious device nodes).

**Figure 8 sensors-22-01672-f008:**
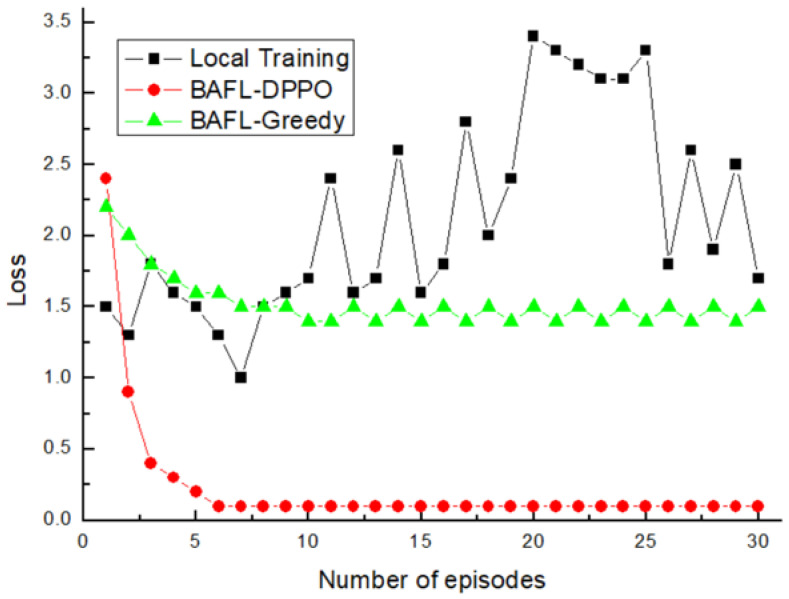
Comparison of losses (30% of malicious device nodes).

**Figure 9 sensors-22-01672-f009:**
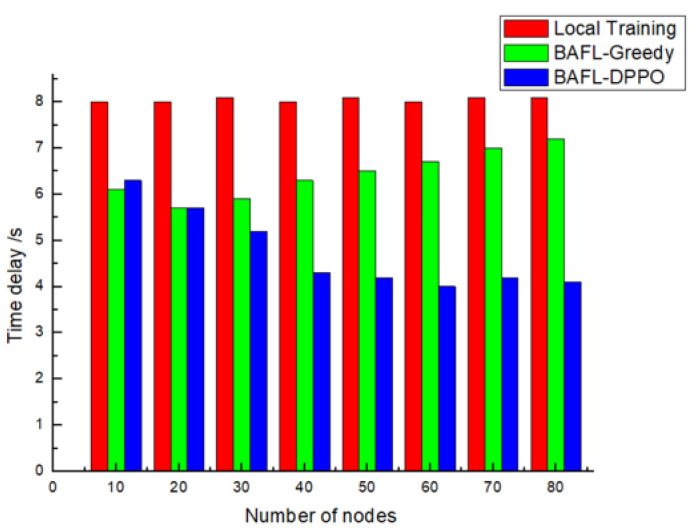
Latency comparison (30% of malicious device nodes).

**Figure 10 sensors-22-01672-f010:**
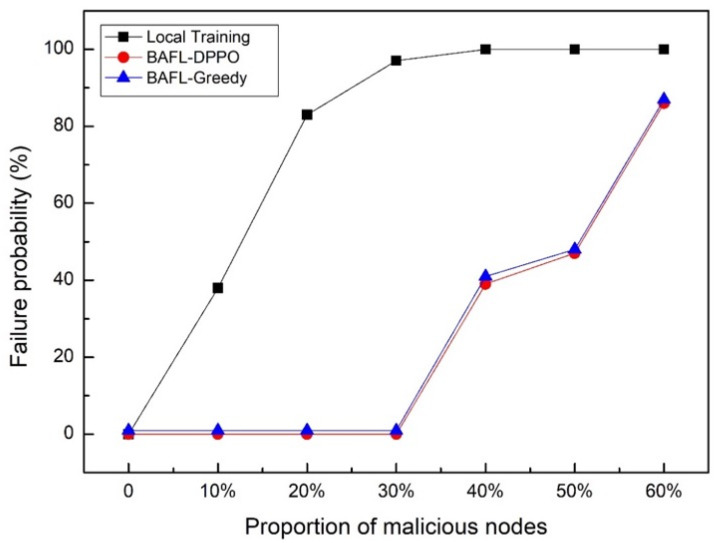
Defending against malicious attack.

## Data Availability

Not applicable.

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
