# Peer review of "Asynchronous Federated Learning System Based on Permissioned Blockchains"

_sensors, 2022, doi:10.3390/s22041672_

Round 1
Reviewer 1 Report
The idea is interesting and publishable. However, the reviewer suggests modifying some parts as below:
- The abstract and conclusion are too long and is not wisely formed. It needs to be revised.
- Since the clarity of the contribution is necessary, please focus on the appliance of this method on applicable large-scale AI/enabled systems for trustful autonomous models and give insight into the introduction's contribution.
- Section 3 is formed as an uncovered plain text; it is better to add subsection wise of the trends of the related work categories. I did not see some recent related works to be covered, like “FED-IIoT: A robust federated malware detection architecture in industrial IoT", which address the same issue in the scope.
- The computational complexity of the proposed algorithms should clearly explored.
- We see the results are given only MNIST dataset which seems the model biased to them, can we have results for another recent dataset to verify this?
Author Response
Dear Editors,
On behalf of my co-authors, we thank you for giving us a chance to revise and improve the quality of our article.
We have read the reviewers’and your comments carefully and have tried our best to revise our manuscript according to the comments. Attached please find the revised version, which we would like to submit for your kind consideration.
We have found an English native speaker with a research background to review our manuscript during revision. And if you think there is any problem, you can raise it at any time. we will look for professional organizations to improve the language.
I wish this revision will be acceptable for publication in your journal.
Thank you for your consideration. I am looking forward to hearing from you.
Sincerely,
Rong Wang
Reviewer 2 Report
The paper presents an asynchronous federated learning system based on permissoned blockchain. The paper is written well. There are some minor revisions need to be incorporated.
- Page 4, line 163 needs to be corrected. (Compared with public blockchains, public blockchains generally have many nodes, and once a blockchain is formed, then the block data cannot be modified, for example, Bitcoin has many nodes, and it is impossible to change the data in it if you want to)
- Please write shorter sentences in order to improve the readability of the paper.
- Algorithm 2, step 4 needs to be improved.
- There are some grammatical mistakes that need to be corrected in next version of your paper.
- Reviewer feels that Section 5 should be more elaborated and justification of simulation results need to extensively presented with proper justification.
Author Response

(The authors gave the same response as above.)

Reviewer 3 Report
On lines 153 and 154, the authors indicated the following:
Blockchain platforms are divided into two categories: public blockchains and private blockchains. Public blockchains are sometimes referred to as not permitted blockchains, while private blockchains are referred to as permissioned blockchains.
I suggest revising this statement because permissionless is for everyone (public), but permissioned can be for everyone (public) or restricted to stakeholders but with permission. Also, I suggest discussing who will control the permissioned blockchain, one entity or a public, Authors should describe the types of blockchain very clearly (public, consortium, and private).
Author Response

(The authors gave the same response as above.)
